

# GenChem v1.0 - a chemical pre-processing and testing system for atmospheric modelling

David Simpson[1,2], Robert Bergström[2,3], Alan Briolat[4], Hannah Imhof[2], John Johansson[2],
Michael Priestley[5], and Alvaro Valdebenito[1]

[1]EMEP MSC-W, Climate Modelling and Air Pollution Division, Norwegian Meteorological Institute, Oslo, Norway
[2]Dept. Space, Earth & Environment, Chalmers University of Technology, Gothenburg, Sweden
[3]Swedish Meteorological and Hydrological Institute, 60176 Norrköping, Sweden
[4]Stockholm Environment Institute at York, Environ. Dept, University of York, Heslington, York, YO10 5DD, United Kingdom
[5]Dept. Chemistry & Molecular Biology, University of Gothenburg, Gothenburg, Sweden

**Correspondence:** David Simpson
(david.simpson@met.no)

**Abstract.** This paper outlines the structure and usage of the GenChem system, which includes a chemical pre-processor (GenChem.py), and a simple box-model (boxChem). GenChem provides scripts and input files for converting chemical equations into differential form for use in atmospheric chemical transport models (CTMs) and/or the boxChem system. Although GenChem is primarily intended for users of the EMEP MSC-W CTM and related systems, boxChem can be run as a stand-
alone chemical solver, enabling for example easy testing of chemical mechanisms against each other. This paper presents an outline of the usage of the GenChem system, explaining input and output files, and presents some examples of usage.

The code needed to run GenChem is released as open-source code under the GNU license.

## 1 Introduction

Atmospheric chemical transport models (CTMs), which simulate the emissions, transport, chemistry and loss processes of
pollutants are essential tools for understanding air quality, and for assisting governments in setting environmental goals and emissions targets. Such CTMs are typically advanced 3-dimensional models with perhaps a million grid-cells. The models account for transport (advection, dispersion) between the cells, and within each cell the chemistry of the atmosphere is simulated, usually with a 'condensed' chemical mechanism (see below), and time-steps ranging from seconds to minutes.

An important CTM in terms of policy is that used by the Meteorological Synthesising Centre-West of the European Monitor-
ing and Evaluation Programme (EMEP MSC-W). The EMEP MSC-W model, described in detail in Simpson et al. (2012) and subsequent articles and EMEP reports (e.g. Stadtler et al. 2018; Simpson et al. 2019 and references therein), is a 3-dimensional Eulerian model whose main aim is to support governments in their efforts to design effective emissions control strategies. The EMEP model has been available as open-source code (www.emep.int) since 2008, and it has since been run by several institutes across Europe (e.g. Solberg et al. 2008; Jeričević et al. 2010; Karl et al. 2014; Omstedt et al. 2015; Vieno et al. 2016; Ots et al.
2018).





As with most CTM systems, the EMEP model code does not directly read chemical equations, but rather requires the production and loss terms of each species to be specified, in a differential form suitable for numerical integration. In order to convert between chemical equations and the numerical form, a chemical pre-processor is used, together with support software, which together comprise the 'GenChem' system.

In addition to the EMEP model, a 1-D model system, the Ecosystem Surface Exchange model (ESX, Simpson and Tuovinen 2014) is being developed as a complement to the EMEP CTM. ESX allows for the investigation of for example chemistry and deposition processes within the lowest tens of meters of the atmosphere (similar in concept to e.g. Makar et al. 1999; Ashworth et al. 2015). The most recent version of the ESX model also allows for Lagrangian trajectory simulations, which will enable the exploration of detailed chemical analysis as air masses traverse perhaps 100s of km (similar to e.g. Hertel et al. 1995; Vieno et al. 2010; Lowe et al. 2011; Murphy et al. 2011). ESX makes use of many components of the EMEP model, including many routines for e.g. radiation, emissions, and the GenChem system.

The most well known chemical pre-processor is probably KPP (Kinetic PreProcessor, Damian et al. 2002; Sandu and Sander 2006), which is used in a number of CTMs (e.g. Ashworth et al. 2015; Eller et al. 2009; Lowe et al. 2009; Langner et al. 1998; Squire et al. 2015; Stroud et al. 2016). KPP is more flexible than GenChem, with for example a range of different chemical solvers available, and with the capability to output as Fortran, C or MATLAB code. GenChem does not aim to compete with KPP in these regards, and in future KPP (or some descendent) may well replace GenChem in the EMEP model system also, but for the purposes of the EMEP CTM, GenChem does have a number of useful features:

1. The GenChem code is tailor-made to produce fortran which can be directly included in the EMEP and ESX systems.

2. The integrated boxChem system allows both direct testing of code prepared for EMEP and side-by-side comparison of chemical schemes.

3. The fortran code produced is more human-readable than with other processors such as KPP, with for example the gas $HNO_3$ being represented by the fortran integer named 'HNO3' instead of by some numeric or abstract variable representation. Similarly, equations in the code produced by GenChem are readily understood, as illustrated in Fig. 1.

4. The code also establishes dry and wet deposition mapping of the chemical species, as well as a number of other characteristics (which can be readily extended through the 'Groups' system, see Sect. 6), such as volatility of organic aerosols or extinction coefficients for aerosol species.

5. The numerical solver, TWOSTEP, is extremely efficient for 3-D chemical transport models, and is thus very apt for the EMEP 3-D CTM or for running complex chemical mechanisms such as the Master Chemical Mechanism (Sect. 5).

6. GenChem has a flexible system that can either calculate molecular weights from chemical formulas, or can use user-provided values.





```
P = rct(40,k) * xnew(MALO2) * xnew(NO)   &
  + rcphot(IDCH3O2H,k) * xnew(MALO2H)   &
  + rcemis(GLYOX,k)

L = rct(48,k) * xnew(OH)   &
  + rcphot(IDCHOCHO_2CO,k)   &
  + rcphot(IDCHOCHO_HCHO,k)   &
  + rcphot(IDCHOCHO_2CHO,k)

xnew(GLYOX) = (xold(GLYOX) + dt2 * P) / (1.0 + dt2 * L)
```

**Figure 1.** Example equations from the output file, CM_Reactions1.txt, for the model species GLYOX. 'P' gives the production terms, and 'L' gives the loss terms, with the last line giving the TWOSTEP solution for this species. See Sect. 1.1, Sect. 7.5

The original GenChem system was written in perl in the 1990s for earlier EMEP model systems, but was converted to a python (2.7) script in 2014. The current structure of the GenChem system, now based entirely on python3, including boxChem, improvements in GenChem.py, and various scripts, was developed between 2016 and 2020.

Although primarily intended for users of the EMEP/ESX systems, the GenChem system can also be run as a stand-alone
chemical solver using the provided "boxChem" driver, enabling for example easy testing of chemical mechanisms against each other. boxChem also provides a useful learning tool for general GenChem usage. This paper outlines the structure of the GenChem system, including boxChem usage and preparation of EMEP-ready model files.

## 1.1   TWOSTEP: chemical equations

The numerical solver used for the chemical equations in EMEP and ESX is the so-called TWOSTEP scheme. This scheme has
been described in detail in Verwer (1994), Verwer et al. (1996) and Verwer and Simpson (1995). It is not the purpose of this article to give details of the TWOSTEP scheme, except to note the simple formulation which results from its use. For example, Fig. 1 illustrates one of the major outputs of GenChem: example code (for the species glyoxal) from the GenChem output file CM_Reactions1.inc. This code is very easy to read, with first the production term (P) which includes time-varying rate cofficients (rct terms), photolysis rates (rcphot) and emissions (rcemis). The loss terms (L) are similar. (Deposition is typically
not included in the loss term, except in some boxChem model applications. The EMEP CTM handles deposition in separate routines, though using variables set in the Species file, see Sect. 6.1.) The 'k' index represents the vertical coordinates used in EMEP/ESX. The TWOSTEP solution is provided in the last line, giving the new estimate for the concentration of GLYOX as a function of the starting concentration (xold), P, L, and TWOSTEP's time-step dt2. In the Gauss-Seidel approach of TWOSTEP these equations are solved one after another for each species in turn, and iterated typically 1–3 times to provide a good solution.
We believe the EMEP model is among the fastest CTMs, and TWOSTEP plays a big role in that efficiency. The inner loop of the TWOSTEP method as applied in the EMEP model makes use only of the chemical concentrations, reaction constants and emission rates for one single gridcell. With order $10^3$ numbers in double precision, approximately 8kB of data, this fits nicely in the lowest level cache in common CPU (L1 cache is typically of size 32kB).





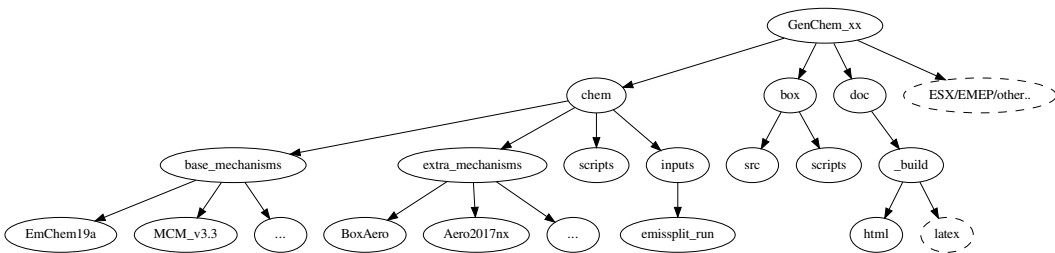

**Figure 2.** Directory structure of GenChem's chem and box directories. The dashed EMEP/ESX directory indicates the possible placement of future EMEP and ESX directory trees in this system.

Sandu et al. (1997) commented that TWOSTEP was by far the best within the class of dedicated explicit methods, and was
an excellent candidate for very large tropospheric gas-phase problems with very small operator split steps. The main limitation
noted by Sandu et al. (1997) was that TWOSTEP is not suitable for aqueous-phase problems, but those are not explicitly treated
within the current EMEP system or the work considered here.

## 2   Installation and code

The code needed to run the GenChem system is released as open-source code under the GNU license, (https://github.com/
metno/genchem), with the user-guide provided at https://genchem.readthedocs.io. GenChem has been developed and tested in
a Linux environment, mainly XUbuntu 16.04–19.10, but has also been tested on Windows systems via a virtual environment.
For those familiar with the Docker system (https://www.docker.com/products/docker-desktop), a Dockerfile and Dockerfile_-
README are provided to enable consistent installation on Windows PCs. See the user-guide for further details.

GenChem is designed to work with modern fortran compilers (tested with gfortran and intel f95), together with python3
($\geq$3.5). As in the EMEP and ESX model systems, double-precision is enforced by compiler options (e.g. -r8 for ifort) rather
than through explicit fortran 'double precision' or 'selected_real_kind' coding. This is partly for aesthetic reasons (we prefer
numbers typed as 1.2 rather than 1.2_wp), and partly for simplicity. Use of these flags ensures that all variables and constants
are automatically elevated to the required precision. Failure to compile with these options will result in an error message. The
testing for this release has mainly been done with the freely available GNU gfortran version 5.
After unpacking the GenChem directory structure should look like Fig. 2. The main technical documentation of this system is
provided at https://genchem.readthedocs.io as noted above, but various markdown-format README.md files are also located
throughout this structure. For example each chemical mechanism should have a README.md file to summarise the mechanism
and any comments.





**Table 1.** GenChem: Inputs and Outputs. For the input emissplit_defaults_voc.csv file and emep_extras the 'BASE' string is replaced by the name of the base chemical mechanism, e.g. EmChem19a.

| Inputs | | Outputs | |
|---|---|---|---|
| File | Section | File | Section |
| GenIn_Species.csv | 6.1 | CM_ChemSpecs.f90 | 7.2 |
| GenIn_Shorthands.txt | 6.2 | CM_ChemGroups.f90 | 7.2 |
| GenIn_Reactions.txt | 6.3 | CM_Reactions1.inc | 7.5 |
| BASE_emissplit_defaults_voc.csv | 6.4 | CM_Reactions2.inc | 7.5 |
| | | CM_Reactions.log | 7.5 |
| | | CM_ChemRates.f90 | 7.6 |
| | | CM_DryDep.inc | 7.4 |
| | | CM_WetDep.inc | 7.4 |
| | | CM_EmisFile.inc | 6.3.2 |
| | | CM_emislist.csv | 6.3.2 |
| | | CM_EmisSpecs.inc | 6.3.2 |
| emep_extras[†]/ | | | |
| BASE_BiomassBurningMapping_FINNv1p5.txt | | CMX_BiomassBurningMapping_FINNv1.5.txt | |
| BASE_BiomassBurningMapping_GFASv1.txt | | CMX_BiomassBurningMapping_GFASv1.txt | |
| BASE_BoundaryConditions.txt | | CMX_BoundaryConditions.txt | |

Notes:† The files in emep_extras are for EMEP model rather than boxChem usage. If intended for EMEP then appropriate fortran code is required. If for boxChem, only dummy files are provided. These files are essential only for the base mechanisms. See the README_emep_extras.md files in the sub-directory.

## 3 Basic usage

**Step 1: initial setup**

GenChem is usually run from a temporary 'work' directory of the box system, e.g. from `tmp_work`. This is created from the box location with:

**Example 3.1.**      `./scripts/box_setup.sh tmp_work`

This will create the working directory, and copy all the files needed for boxChem into it.

**Step 2: do.testChems**

Once setup, the user is ready to build and run some chemical schemes. With the example of EmChem19a, and now from our `tmp_work` directory, the simplest next step is:

**Example 3.2.**      `./do.testChems EmChem19a`





This script will run GenChem.py on the EmChem19a scheme (also adding a few extra_mechanism files as discussed in Sect. 5), run 'make', and then run the resulting box-model code. Results will appear in one log file (e.g. RES.EmChem19a), and as comma-separated results in the Outputs directory (OUTPUTS): file boxEmChem19a.csv. This file is readable with e.g. libreoffice. Plot scripts are also available for easy visualisation and comparison of these csv results (Sect. 4.2).

The 'CM_' and 'CMX_' fortran files produced by this process are saved in directories, e.g. here in 'ZCMBOX_EmChem19a'. Now, if one wants to compare several schemes, one can do e.g.:

**Example 3.3.**       *./do.testChems EmChem19a CB6r2Em CRIv2R5Em*

This would produce 3 output .csv files, which again are easily plotted against each other (see Sect. 4.2).

For completeness, we can mention that it is possible to run the simpler script do.GenChem from the same directory (or indeed from elsewhere), e.g.:

**Example 3.4.**       *./do.GenChem –b EmChem19a*

This will run the GenChem.py script, and generate CM and CMX files, but it will not attempt to compile or run boxChem.

**Step 3: emep_setup.py**

The do.testChems script described above is best for quickly testing and comparing different mechanisms. Usually these comparisons only involve gas-phase mechanisms such as EmChem19a or MCMv3.3Em. However, the EMEP model usually requires a host of extra species and reactions to accommodate secondary inorganic aerosol, sea-salt, dust, organic aerosols, and pollen, as discussed in Sect. 5. The EMEP CTM also requires files to specify how emissions and boundary conditions should be distributed among specific species, e.g. how a VOC emission should be split into C2H6, C2H4, nC4H10 etc.

In fact, GenChem produces many files which are copied into the appropriate ZCM_ directories, e.g. ZCM_EmChem19a-vbs for the scheme EmChem19a-vbs, as indicated in Table 1. The recommended way to get this directory is to use the script 'emep_setup.py' from your temporary work directory within the box system. So, from e.g. box/tmp_work, do:

**Example 3.5.**       *./emep_setup.py EmChem19a-vbs*

or just::

**Example 3.6.**       *./emep_setup.py*

and this will provide a list of available options. Users can easily edit the emep_setup.py scripts to modify the extra_-mechanisms used – see Sect. S2 in the Supplementary Information.

**Step 4: from box to EMEP CTM**

After emep_setup.py has successfully run, the ZCM_ directory produced contains all the files needed to run the EMEP CTM. The CM_ and CMX_ files can be copied directly to the CTM's source directory, and the EMEP model compiled as normal (make clean, make). The emissplit_run files need to be sent to a location specified by the user (via the EMEP CTMs' emep_-config.nml namelist).





## 4   boxChem - GenChem's box-model

As noted in Sect. 3, boxChem is an integral part of the GenChem system. boxChem provides a way of testing GenChem implementations and is indeed strongly recommended as the main method of preparing ESX and/or EMEP codes. As a stand-alone model, boxChem is also a valuable way to compare results from different chemical mechanisms.

The usual procedure is to use 'do.testChems' as was illustrated in Sect. 3. Running do.testChems will run GenChem.py and copy all necessary CM-files to the user's work directory. Input variables like temperature, relative humidity or anthropogenic emissions stay constant over the simulation time, as set in config_box.nml. Photolysis rates, however, change every time-step according to the sun's zenith angle. Biogenic emissions may also be modified by zenith-angle, if the simple SUN variable is used (see Sect. 6.3.2).

### 4.1   config_box.nml

The namelist input file config_box.nml allows the user to conrol many aspects of the boxChem model run. This file specifies the start and end time as well as the time step (dt) to be used. If emissions are wanted, the logical variable 'use_emis' has to be set on 'T' (=true), and the anthropogenic emissions given as simple totals (e.g. 'nox', 18.3 uses an emission of NOx of 18.3 kg m$^{-2}$ day$^{-1}$). boxChem makes use of default speciations of compound emissions such as NOx or VOC – see Sect. (S2.3) in Supplementary Information for more information on these splits and how they can be changed.

The concentration of the fixed species M and H2O, and initial concentrations of all species, are also set in config_box.nml. M and H2O can either be set directly in molecules cm$^{-3}$ or by defining the pressure and relative humidity, respectively.

The OutSpecs_list variable in config_box.nml specifies which pollutants are required in the output file, though by default it is set to 'all'. This output file, e.g. box_dt30s.csv (where 30s is the 'external' timestep used), is generated in the outputs directory (e.g box/tmp_work/OUTPUTS), and gives hourly values for each species specified in OutSpecs_list, along with appropriate units. For gaseous species we use ppb or molecules cm$^{-3}$, and for particulate matter $\mu$g m$^{-3}$.

The choice of time-step is discussed in the Supplementary Information. See the comments in config_box.nml for further details about boxChem and config_box.nml setup and usage.

### 4.2   Plotting and examples

The python/matplotlib script boxplots.py (found in the box/scripts directory) can plot either individual or multiple species produced by boxChem, and for one or several output files. For example, if one has run say two chemical schemes using Step 2 above, the results are easily plotted from the `box/tmp_work/OUTPUTS` directory:

**Example 4.1.**         *../../scripts/boxplots.py -h for help!*

**Example 4.2.**         *../../scripts/boxplots.py -v O3 -i boxEmChem19a.csv boxChemX.csv -p*

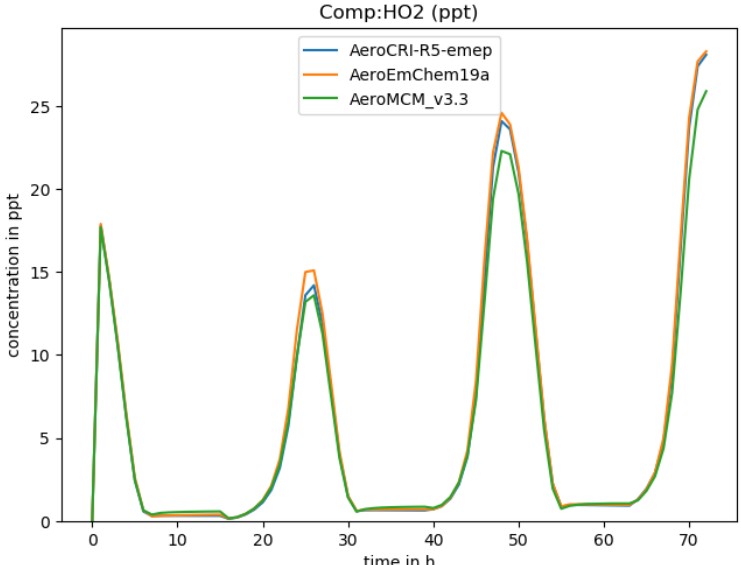

**Figure 3.** Example of comparison of three chemical schemes, produced for HO2 with the boxplots.py script

where -v gives compound to be plotted, and -p produces a png graphics file as well as screen output. Using 'ALL' or 'DEF'
with -v results in all or many common species being plotted at once (-p is assumed in this case). For example, Fig. 3 shows a
comparison of three schemes produced with this script.

Another helpful script just grabs the concentrations:

**Example 4.3.** `../../scripts/getboxconcs.py O3 boxEmChem19a.csv`

which results in ResConcs_boxEmChem19a_O3_ppb.txt

**5 Chemical mechanisms**

We provide a number of chemical mechanisms which have been formatted for GenChem usage. These mechanisms are organ-
ised into two types, with separate directory trees:

– 'base_mechanisms'

These schemes are typically fairly complete sets of gas-phase photochemical mechanisms, and are designed to be the
175 core for any boxChem, ESX or EMEP runs. Apart from the EMEP-developed EmChem19a, the other base-schemes have
been adapted from other sources for EMEP usage, hence the 'Em' postfix. Details of these schemes and adaptations can
be found in Bergström et al. (2020a). The schemes provided with GenChem currently comprise (see also Table 2):





- EmChem19a — the base EMEP chemical scheme, which has 158 gas-phase reactions in the core mechanism, and in addition to these a number of heterogeneous reactions are also included, bring the total to 171 reactions for simple boxChem usage (c.f. Table 2). This scheme is a surrogate-species scheme that has evolved over many years (Eliassen et al., 1982; Simpson et al., 1993, 2012; Bergström et al., 2020a), and has over the years been shown to compare well against other and more extensive chemical mechanisms (Kuhn et al., 1998; Andersson-Sköld and Simpson, 1999; Bergström et al., 2020a). The most recent changes have included a revised isoprene chemistry based on the CheT2 mechanism of Squire et al. (2015), and the addition of toluene and benzene as well as o-xylene to represent aromatics. A new feature of EmChem19a compared to earlier EMEP schemes is the addition of an RO2POOL species, representing the total concentration of all peroxy radicals; RO2POOL is used for setting the rates of peroxy + peroxy radical reactions. A set of new nitrate radical reactions has also been added, and reaction rates have been revised to be in line recent IUPAC recommendations. For details see Bergström et al. (2020a).

- MCMv3.3Em — based on the 'Master Chemical Mechanism' v3.3.1, with a few updated reactions (mainly updates of some reaction rates to be in agreement with IUPAC recommendations 2009–2018). In our implementation the MCM mechanism has over 5800 species and over 17000 reactions. See Jenkin et al. (2015), and references therein, for details about MCM and Bergström et al. (2020a) for details about the revisions made for MCMv3.3Em. The MCM mechanism is too large for the EMEP CTM, but can be run with boxChem or ESX, and serves as an important reference mechanism.

- CRIv2R5Em is an adaption of the 'Common Representative Intermediates' scheme, with a variant of the CRI v2.2 isoprene chemistry (Jenkin et al. 2008, 2019). In order to make the scheme manageable for 3D-modelling the full CRI scheme is reduced by only including emissions from a limited set of different VOCs (the so called CRI_R5 reduction subset from Watson et al. 2008 is used in the EMEP adaption of CRI). Even with this reduction the CRI scheme is substantially larger than our EmChem schemes, but still well suited to 3-D modelling (see e.g. McFiggans et al., 2019 and Jenkin et al., 2019, for studies employing the CRI-R5 mechanism with the 3-D EMEP MSC-W model). The EMEP version of CRI v2-R5 (CRIv2R5Em) is described in detail by Bergström et al. (2020a) and the revision of the isoprene chemistry by Jenkin et al. (2019).

- CB6r2Em — The 'carbon-bond' (CB) schemes have been developed over many years as an innovative solution for dealing with chemistry in 3-D CTMs (Gery et al., 1989; Yarwood et al., 2010a, b; Luecken et al., 2019). The CB6r2 chemical scheme has been implemented without any significant change in the GenChem, except that photolysis rates have been adjusted to use MCM (for boxChem usage) or EMEP CTM surrogates. Also, the biogenic VOCs of CB6r2, ISOP (isoprene) and TERP (representing all monoterpenes, MT), have been renamed to the C5H8 and APINENE (also a surrogate for all MT in this case), since this allows the same emission reaction equation to be used for all four mechanisms if desired.

- 'extra_mechanisms'





In this directory we store sets of reactions and sometimes species that can be appended to the base mechanisms. Many of these are essential for 3-D chemical transport modelling, whilst others are used for box model simulations. With this release we provide mechanisms for sea-salt, dust, emissions from ships (EMEP uses a special ShipNOx species, see Simpson et al. 2015), and several organic aerosol and BVOC emission options. Comments on each scheme can be found in the appropriate README files. The organic aerosol schemes are further discussed and compared in **?**.

Tables S1–S2 in the Supplementary information provide brief explanations of the many currently implemented extra_mechanism packages, but we can give three important extra mechanisms as examples:

– BoxAero

Provides $SO_2$ gas phase chemistry and some reactions for very simple gas to aerosol conversion for $SO_3$, $HNO_3$ and $N_2O_5$. The reactions provide simplified chemical loss mechanisms for these species in the box model — they are calculated in a more complex way in the full EMEP model, which also includes $NH_3$ chemistry. This directory is intended only for boxChem usage, and is applied automatically when using the do.testChems script (see Sec.3).

– PM_WoodFFuelInert

PM emissions (fine and coarse) in the EMEP model are typically split into EC (elemental carbon), POM (primary particulate organic matter) and remPPM components. Different levels of detail are allowed, but this package enables the most common setup. POM and EC emissions are divided into those from biomass combustion ("wood-burning") and fossil-fuel. POM are assumed inert, consistent with the 'NPAS' PM_VBS_EmChem19 scheme discussed below. EC emissions are further divided into 'new' and 'age' components, to reflect the level of hydrophobicity (Tsyro et al., 2007; Genberg et al., 2013) In some inventories primary sulphate is also provided, represented here as pSO4. remPPM represents all PM components which are not explicitly specified from these other compounds.

– PM_VBS_EmChem19

Provides additional organic aerosol reactions for EmChem19a. These reactions are currently (in version rv4.34) default in the EMEP CTM, and represent minor updates of the VBS schemes presented in Bergström et al. (2012, 2014); Simpson et al. (2012). The default scheme used in PM_VBS_EmChem19 uses the 'NPAS' version of the EMEP VBS mechanisms (with inert primary emissions and aging of SOA compounds) – see Simpson et al. (2012) (and its SI) for further details. Unlike the simple gas-phase compounds used elsewhere, SOA species are tracked as a true aerosol — with one compound representing the sum of gas and particle-phase compounds. These semi-volatile species and reaction formats are discussed more in Sect. 6.1.1 and Sect. 6.3.

Table 2 summarise the number of species and reactions involved in typical boxChem or EMEP CTM usage, and Tables 3–4 gives examples of the combinations of base_ and extra_mechanisms packages. It can be seen that the GenChem system allows a very flexible approach to explore different levels of chemical complexity, especially for EMEP CTM applications. Both base and extra mechanisms will be expanded in future GenChem versions, for example with further organic aerosol modules.





**Table 2.** Comparison of chemical mechanisms provided with GenChem 1.0, in either boxChem (mainly gas-phase) or EMEP CTM (with many particle and semivolatile compounds and tracers) configurations: number of species ($N_s$), number of rate-coefficients, ($N_r$, includes photolysis), number of photolysis reactions ($N_j$), and number of anthropogenic emission terms ($N_e$).

| | boxChem[a] | | | | EMEP CTM[b] | | | |
|---|---|---|---|---|---|---|---|---|
| | $N_s$ | $N_r$ | $N_j$ | $N_e$ | $N_s$ | $N_r$ | $N_j$ | $N_e$ |
| EmChem19a | 80 | 171 | 34 | 21 | 127 | 198 | 34 | 48 |
| CB6r2Em | 127 | 227 | 27 | 30 | 174 | 252 | 27 | 57 |
| CRIv2R5Em | 225 | 575 | 111 | 35 | 272 | 602 | 111 | 62 |
| MCMv3.3Em[c] | 5842 | 17220 | 3120 | 141 | | | | |

Notes: (a) EmChem19a-box, CB6r2Em-box, CRIv2R5Em-box, cf. Table 3 (b) EmChem19a-vbs, CB6r2Em-vbs, CRIv2R5Em-vbs, cf. Table 4 (c) Unlike the other provided schemes, MCMv3.3Em includes many halogen reactions. These are included for future developments. Further, MCM treats all individual reactions paths as separate reactions, whereas the other schemes frequently combine reactions into a single net reaction.

**Table 3.** Examples of Base and extra mechanisms associated with boxChem configurations.

| Label | base_mechanism | extra_mechanisms | Comment |
|---|---|---|---|
| EmChem19a-box | EmChem19a | BoxAero, BoxDep, BVOCemis | boxChem schemes all use same extras |
| CB6r2Em-box | CB6r2Em | as above | |
| CRIv2R5Em-box | CRIv2R5Em | as above | |
| MCMv3.3Em-box | MCMv3.3Em | as above | |





**Table 4.** Examples of Base and extra mechanisms associated with EMEP CTM configurations (via emep_setup.py).

| Label | base_mechanism | extra_mechanisms | Comment |
|---|---|---|---|
| EmChem19a-vbs (or EmChem19a)[a] | EmChem19a | PM_VBS_EmChem19[b], BVOC_IsoMT1_emis, +COMMON[c] | Standard EMEP, VBS SOA for $\alpha$-pinene surrogate |
| EmChem19p-vbs (or EmChem19p)[a] | EmChem19a | As EmChem19a-vbs, + Pollen | Open-source EMEP has pollen |
| EmChem19a-vbs3 | EmChem19a | As EmChem19a-vbs, + BVOC_ExtraMTs, PM_VBS_ExtraMTs, BVOC_IsoMT3_emis, +COMMON[c] | with 3 mono-terpenes |
| CB6r2Em-vbs | CB6r2Em | PM_VBS_CB6r2Em, BVOC_IsoMT1_emis, +COMMON[c] | CB6 gas-phase, VBS SOA for $\alpha$-pinene surrogate |
| CB6r2Em-H | CB6r2Em | PM_Hodzic_CB6, BVOC_IsoMT1_emis, +COMMON[c] | CB6 gas-phase, Hodzic-like SOA[d] |
| CRIv2R5Em-vbs | CRIv2R5Em | PM_VBS_EmChem19, BVOC_IsoMT1_emis, +COMMON[c] | CRI gas-phase, VBS SOA for $\alpha$-pinene surrogate |
| CRIv2R5Em-M19 | CRIv2R5Em | BVOC_XTERP_CRI, PM_Hodzic_Aromatics, PM_JPAC_MT3, BVOC_IsoMT3_emis, +COMMON[c] | SOA as used in McFiggans et al. (2019) |

Notes: [a] The simpler terms EmChem19a and EmChem19p are used in EMEP CTM rv4.34 (current at time of writing). [b] EMEP's default 'PNAS' scheme - see Sect. 5 and SI Table S2. [c] COMMON=Aqueous, Aero2017nx, ShipNOx, PM_FFireInert, SeaSalt, DustExtended, Ash, PM_WoodFFuelInert, BVOC_SQT_NV. See SI Table S1 for further explanation of these packages. [d] Loosely based upon Hodzic et al. (2016).




**Table 5.** Example content lines from GenIn_Species.csv input file .

| Spec | Type | Formula | MW | DRY | WET | Groups | !Comments |
|------|------|---------|-----|-----|-----|--------|-----------|
| OD | 0 | O | xx | xx | xx | xx | ! |
| OH | 0 | OH | xx | xx | xx | xx | ! |
| O3 | 1 | O3 | xx | O3 | xx | OX | ! |
| NO2 | 1 | NO2 | xx | NO2 | xx | NOX;OX;OXN;daObs | ! |
| HCHO | 1 | HCHO | xx | HCHO | HCHO | xx | ! |
| CH3O2 | 1 | CH3O2 | xx | xx | xx | RO2 | ! |
| C2H5OOH | 1 | C2H5OOH | xx | ROOH | xx | xx | ! |
| C5H8 | 1 | C5H8 | xx | xx | xx | BVOC | ! |
| BVOCNO3 | 1 | someNO3 | 199.99 | xx | xx | nitrate;OXN | ! demo species |
| C918NO3 | 1 | C9H13NO5 | 215.2032 | xx | xx | xx | ! |
| RO2POOL | 1 | special | xx | xx | xx | xx | ! Sum of RO2 |
| BSOC_ng1e2 | 2 | C | 12 | ALD | ROOH | CSTAR:0.1;DeltaH:30; OM25;PCM;BSOA | ! |
| EC_f_wood_new | 3 | C | xx | PMf | ECfn | Extinc:ECn;EC_f;PMfine;... | !Primary wood burning EC$_{2.5}$ |

Note: commas and comment lines from file omitted for clarity.

# 6 Input files

The three input files to GenChem.py are GenIn_Species.csv, GenIn_Shorthands.txt and GenIn_Reactions.txt. In addition, a
245 mechanism-specific emissplit file is needed in order to tell models how VOC emissions are to be split into individual compounds. These files are discussed below in Sect.6.1–6.4.

## 6.1 GenIn_Species.csv

The GenIn_Species.csv file is a spreadsheet-friendly comma-separated file where the characteristics of the chemical compounds are given. Table 5 gives some example entries, and we discuss each column below.

**Spec** Species name as used in model.

**Type** Type of compound. CTMs usually distinguish between advected and non-advected (or short-lived) species, in order to minimise CPU needs (concentrations of short-lived compounds only need chemical reaction terms, not advection). In addition, the EMEP model handles semivolatile SOA species through special handling (Sect. 6.1.1), and some species are so chemically unreactive (e.g. EC_f_wood_new) that they can be accurately calculated without multiple iterations.
Allowed values of type are: 0 – for short lived (e.g. OH) , 1 – for advected (e.g. O3, HCHO), 2 – for semivolatile SOA compounds (e.g. BSOC_ng100), and 3 – for compounds which react very slowly (e.g. CH4).





**Formula** If a true chemical formula is provided (e.g. CH3CHO, or O=CHC(O2)(CH3)CH2OH) then GenChem will calculate the number of atoms (C, H, O, S or N) and the molecular weight. Such formula must use capital letters; lower case letters are ignored as far as processing is concerned, but may be used to help document the intention, e.g. nC4H10 is identical to C4H10, or pm25 is particulate matter but whose formula we do not know. The entry for BVOCNO3 in Table 5 has formula 'someNO3' which mixes lower and upper case. In this case the molecular weight must be given if this is needed for the chemical modelling. (Typically we do need the mass of emitted and particulate species, but not always the mass of other gas phase species since number concentrations are used in the chemical solver, mixing ratios are used for advection, and output is often in ppb units. Occasionally examples occur where mass is not strictly required, but where one wants to know the nitrogen content, typically where outputs are given in terms of e.g. $\mu g(N)/m^3$. In this case, the 'someNO3' formula would be enough to allow GenChem to figure out that this compound contains one nitrogen atom.)

**MW** Can be blank (xx) or a real number giving the molecular weight of the compound. When given, this value is used in place of any MW calculated from the formula. As noted above, the MW value is sometimes, but not always, needed. For some emitted compounds, usually connected with particulate matter where we do not know the composition, we have to give a dummy molecular weight. This information is used internally in the model to get associated mixing ratios, but outputs for such compounds should always be in mass-units so that consistency is preserved.

**DRY** Dry-deposition surrogate. The EMEP and ESX models calculate dry-deposition explicitly for a limited number of compounds, and here we can choose which of these compounds can be used as a surrogate for the desired species. For example, for O3 we simply use O3; for C2H5OOH we use the MEOOH surrogate. If not dry-deposited, simply use xx. Note that for the semivolatile SOA species EMEP/ESX CTMs will use this rate for the gas-phase fraction of the SOA; deposition of the particle phase is treated using the EMEP standard parameters for fine particles.

**WET** Wet-deposition surrogate - similar to the dry deposition system. For example, for HCHO we simply use HCHO; for the semivolatile SOA species, such as BSOC_ng1e2, we specify the wet-deposition surrogate for the gas-phase fraction of the SOA (e.g. ROOH in the example in Table 5); for the particle fraction the standard wet deposition parameters for fine-particulate matter is used automatically.

**Groups** Specifies groups which the species belong to (e.g. OXN for oxidised nitrogen, RO2 for peroxy radicals) and allows surrogate species or factors to be assigned to these groups, e.g. Cstar:10.0;Extinc:0.4. It is important that these groups are separated by semi-colons, not commas. This rather powerful feature is discussed further in Sect. 6.1.2.

**Comments** Can be anything; this is not used by GenChem.

### 6.1.1 Semivolatile SOA species

Type 2 signifies secondary organic aerosol (SOA) species. These are also subject to advection, but in addition they are semivolatile and exist in both gas and particulate phase. The EMEP model tracks such species by compound rather than phase, and calculates the partitioning between the phases dynamically, based upon the compound's volatility (Bergström et al., 2012;





Simpson et al., 2012). The approach used, the volatilty-basis-set (VBS) follows methods developed by Donahue et al. (2006),

Robinson et al. (2007) and colleagues — see Bergström et al. (2012) and references therein for details. For GenChem purposes, species labelled with type 2 are accounted within the list of advected species, but the start and end of the semivolatile list is calculated by GenChem.py, to produce integer variables which demarcate these semivolatile compounds, e.g. `FIRST_SEMIVOL=136` and `LAST_SEMIVOL=176`. (Note, GenChem will reorder different types of species to be consecutive, so despite the order of species in the GenIn_Species.csv file, all type-2 species will lie together in the index range

`FIRST_SEMIVOL=136:LAST_SEMIVOL=176`.)

Type 2 (SOA) species require specification of their effective saturation concentrations ($C^*$) and the enthalpies of vaporization ($\Delta H_{vap}$), following normal VBS principles. These specifications are made using the 'Group' methods described next.

### 6.1.2 Groups

The 'Groups' mechanism plays an important role for feeding key information to the EMEP/ESX models. Some of them are

300 indeed essential — for example, in EmChem19a, CRIv2R5Em and MCMv3.3Em, the RO2_GROUP needs to be set correctly to get the correct concentrations of the special RO2POOL concentration, and the deposition of groups of compounds (e.g. oxidised nitrogen deposition) depends on those compounds being correctly identified by their groups. Groups have to be separated by a semicolon, and there are two types of group labels for a specific species:

(i) simple name, e.g. OXN — indicates that this species that belong to a group, in this case OXN for oxidised nitrogen

compounds. In the CTM code, the members of one group are easily accessible so they can be treated specially (see Sect. 7.3).

(ii) compound groups which specify numerical or character values to pass specific properties, e.g. the groups CSTAR:0.1 and DeltaH:30 for BSOC_ng100, or Extinc:ECn for EC_f_wood_new in Table 5.

If a species is both a member of a type (i) group and has a (wet or dry) deposition surrogate, additional WDEP or DDEP

groups will be automatically generated, e.g. `DDEP_OXN_GROUP`.

The specification of numerical or character values (group type (ii)) is indicated with a colon notation (as opposed to the semicolon used to separate groups). For example, SOA species which use the VBS system require specification of their effective saturation concentrations at 298K ($C^*$, in $\mu$g m$^{-3}$) and the vaporization enthalpies ($\Delta H_{vap}$, in kJ mol$^{-1}$), or for aerosol optical absorption we need extinction coefficients. These specifications are simply set through the groupings Cstar, deltaH and Extinc,

with e.g. Cstar:1.0e-2;deltaH:30.0 for SOA, and Extinc:SO4 for sulphate. Sect. 7.3 provides further explanation of such groups.

These groupings are not hard-coded in GenChem, and may or may not be used by any CTM, so this system provides an easily extensible mechanism for introducing new characteristics into modelling systems.

### 6.2 GenIn_Shorthands.txt

Rate-coefficients (Sec. 6.3) in chemical equations often make use of shorthand notation. For example, we might use **KHO2RO2**

as a generic rate coefficient for HO2 + RO2 reactions. In the EMEP/ESX system some variables have special names, and these





```
FH2O                (1.0+1.4e−21*h2o*exp(2200*TINV))
*
K80                 3.2e−30*M*(TEMP/300)**(−4.5)
K8I                 3.0e−11
KR8                 K80/K8I
FC8                 0.41
NC8                 0.75−1.27*(LOG10(FC8))
F8                  10**(LOG10(FC8)/(1+(LOG10(KR8)/NC8**2)))
KMT08               (K80*K8I)*F8/(K80+K8I)
KMT3_OH_HNO3        KMT3(2.4e−14,460.,6.5e−34,1335.,2.7e−17,2199.)
```

**Figure 4.** Selected lines from the input file, GenIn_Shorthands.txt

```
TROE_OH_NO           : OH + NO = HONO ; Ref1
1.44e−13+M*3.43e−33  : [OH]  + CO_FIRE  =   ;  tracer
2.2e−10              : OD + <H2O> = 2. OH      ; A97,J
TROE_NO_OP           : OP + NO  + {M}   = NO2    ; A97,J
1.36e−11             : [OXYL] + [OH] = |YCOXY(0)|  ASOC_ug1  + ...
```

**Figure 5.** Selected lines from the input file, GenIn_Reactions.txt

are understood by GenChem. The variables which are known to EMEP/GenChem are: TEMP (temperature), TINV (inverse temperature), M, O2, N2 and H2O (concentrations of air, oxygen, nitrogen and water vapour).

The name and the expression for a shorthand have to be separated by whitespace for GenChem to process it. Names of species can also be used in shorthands expressions. Figure 4 illustrates several examples, including how short-hands defined

earlier can be re-used in the same system — as done for the MCM example in producing the KMT08 variable. The last example, for KMT3, also illustrates that the right-hand side can be a function, which of course needs to be compatible with the fortran code of the calling code.

### 6.3 GenIn_Reactions.txt

The GenIn_Reactions.txt file contains the chemical reactions, with format:

```
rate coefficient : reaction ; [optional comment]
```

and with the reaction consisting of reactants and products, along with their stoichiometric factors as appropriate. The semi-colon marks the end of the reactions, and whitespace is needed between all terms, e.g. between a stoichiometric factor and a

species. Some typical lines are given in Fig. 5. The first line here is trivial, in the sense that OH, NO, and HONO are all normal chemical species as defined in GenIn_Species.csv, and GenChem will add production and loss terms appropriately for each, with a reaction rate given by the TROE_OH_NO shorthand.





GenChem is flexible as to whether products are written explicitly or with stoichiometric coefficients (i.e. 2 OH is the same as OH + OH). Non integer stoichiometric coefficients are allowed, since we often condense multiple branches of a reaction into one equation for CTM use.

### 6.3.1 The funny brackets: tracers, explicit and dummy catalysts, and yield modifiers ([], <>, {} and ||)

Four types of brackets are used in GenChem_Reactions files to modify the way compounds or yields are handled:

**Case [ species ]:** The second reaction of Fig. 5 illustrates a nice feature of GenChem; tracers are easily added. Here we have a tracer of biomass burning CO, which we can track in the EMEP model. This CO_FIRE tracer is emitted along with the 'real' CO, but its existence is not allowed to affect the standard photochemistry. The [] around the OH signifies that this CO tracer has a chemical loss due to OH, but that we do not allow the model's OH to be degraded by this artificial species. The lack of products in this example also signifies that we do not track the products of this loss, just the CO_FIRE itself.

**Case < species >:** The third example from Fig. 5 illustrates the use of <> notation. In this case, the species within the angle-brackets is not one of the chemical compounds tracked by the CTM, but rather some compound whose concentration is effectively constant and known to the EMEP/ESX system. The compounds used so far in EMEP/ESX are H2O, N2, O2 and M (air molecules). This last line could equivalently have been written:

**Example 6.1.**      `2.2e-10*H2O : OD = 2. OH ;`

where the H2O concentration now applies as a simple part of the rate coefficient. (This is actually exactly the way GenChem handles this internally).

**Case { species } :** The fourth reaction in Fig. 5 shows another type of special notation. Species within curly brackets are not used in any way, but they can be added to the reactions as comments illustrating reactants, whose concentrations are already included in the rate expression — in the example the TROE_NO_OP rate expression takes into account the pressure (i.e. [M]) dependence for this 3-body reaction.

**Case | yield-expression |:** The final reaction of Fig. 5 illustrates the use of || brackets. These are sometimes used in the reaction schemes for secondary organic aerosol, as seen here for the production of the semivolatile ASOC_ug1 species. The contents of || represent yield coefficients which will be updated each time-step in the EMEP model. The output CM_Reactions2.inc file for this case includes the term:

**Example 6.2.**      `P = YCOXY(0) * 1.36e-11 * xnew(OXYL) * xnew(OH)`

These variables (here YCOXY(0)) must be predefined in order for emep_setup.py and the emep model to compile.





```
emisfiles:sox,nox,co,voc,nh3
*
rcemis(NO,KDIM)        : = NO     ;
rcemis(NO2,KDIM)       : = NO2    ;
rcemis(NC4H10,KDIM)    : = NC4H10    ;

rcphot(IDO3_O1D)       : O3            = OD         ;    J(5e-5)
rcphot(IDH2O2)         : H2O2          = 2 OH       ;    J(8e-6)
```

**Figure 6.** Emissions and photolysis lines from the input file, GenIn_Reactions.txt

### 6.3.2    Emissions

When using emissions in GenIn_Reactions.txt, the labels used for associated emission files have to be defined in a special line, e.g. 'emisfiles:sox,nox,co,voc,nh3' as in the first line of Fig. 6. These emission labels (e.g. nox) are those used in EMEP/ESX for emission input files, and also the file endings for the respective emissplit file, see Sect. 6.4. Other labels can easily be used
and defined, as long as the emissplit system exists to convert these groups into model species (e.g. nox into NO and NO2, or voc into C2H6, NC4H10, etc.).

The next three lines in Fig. 6 are examples for emission reactions: the reaction rate is denoted as rcemis(SPECIES,KDIM) and there is no reactant in the reaction. GenChem will replace KDIM with the vertical cordinate, assumed to be k, in the fortran code, e.g. giving rcemis(NO,k).

Biogenic VOC (BVOC) emissions are special, in that specific functions exist in the EMEP model for dealing with the light, temperature, and other dependencies of these. In the extra_mechanisms files used for the EMEP CTM setup, we use the 'rcbio' functions as shown in Fig. 7.

When using boxChem, a very simplified system is used for BVOC emissions, illustrated also in Fig. 7. The 'SUN' variable (borrowed from the KPP system) allows for simple variation of emissions with zenith angle (and gives zero emissions at night).
The numerical coefficients (5.0e7 or 2.5e6) correspond to typical emission rates (see the appropriate mechanism file), and the fIsop, fMTL and fMTP factors provide a scaling factor which can modified in config_box.nml.

### 6.3.3    Photolysis

The reaction rates of photolysis reactions are denoted as rcphot(PHOT_ID). GenChem will automatically add the 'k' dependency on the vertical level to the photolysis rate (e.g. to give rcphot(IDH2O2,k)). The index variables (e.g. IDO3_O1D)
refer to photolysis rates as defined in the EMEP/ESX/boxChem codes.

### 6.4    emissplit files

Emissions are often provided to models for groups of compounds, e.g. NOx for NO & NO$_2$, and NMVOC for non-methane volatile organic compounds. These emissions need to be assigned to individual chemical compounds, and converted from mass to number using the appropriate molecular weight.





```
boxChem simplified biogenic emission rates:

5.0e7*SUN*fIsop : =  C5H8   ;
2.5e6*SUN*fMTL  : =  APINENE  ;
2.5e6*fMTP      : =  APINENE  ;

Typical EMEP biogenic emission rate:
*
_func_rcbio(1,k) : =  C5H8 ;
_func_rcbio(2,k) : =  APINENE ;
```

**Figure 7.** Biogenic emissions lines from input files used in GenIn_Reactions, for either EMEP or boxChem setups

The default files for sox, nox, co and nh3 are identical across all provided schemes, and provided in the input/emissplit_defaults directory as files such as emissplit_defaults_nox.csv. For NMVOC and PM inventories specific files are needed for each chemical mechanism, and sometimes depending on available inventories.

Default NMVOC emission splits are provided in GenChem for 11 different source categories (covering traffic, agriculture, etc), according to the so called 'SNAP' classifications. The provided data are based upon average UK emission profiles from
Passant (2002) and emissions from 2010, and have been adapted in this work for each base chemistry scheme.

For GenChem we provide such NMVOC files for all supported chemical mechanisms, in the appropriate directory. Thus, for EmChem19a the file for NMVOC emissions would be named EmChem19a_emissplit_defaults_voc.csv. For boxChem testing the do.GenChem or do.testChems scripts will move this file to inputs/emissplit_run and also copied to ZCMBOX_EmChem19a/emissplit_run and rename to simple emissplit_defaults_voc.csv. If emep_setup.py is used, the emis-
split_run dircetory is copied to ZCM_EmChem19a/emissplit_run.

In the EMEP CTM it is common for these default values to be overridden by 'emissplit_specials' files which can assign country and sector-specific NOx, NMVOC and PM profiles. Such profiles need to be generated by the EMEP CTM user, however, and are not part of GenChem. An example of this system, and such emission splits, is given in the Supplementary Information, Table S5, of Simpson et al. (2012).

However, in boxChem, users may of course modify any of these emissplit files - see Sect. S2.3 in Supplementary information.

## 7   Output files

The output files, prefixed with CM_ to denote chemical mechanism, are summarised in Table 1, and discussed in the relevant section below.

### 7.1   CM_ChemDims_mod.f90

The CM_ChemDims module provides basic information (Fig.8) about the dimensioning of the chemical system, giving for example the total number of species (NSPEC_TOT), photolysis rates (NPHOTOLRATES) or emission files (NEMIS_File).





```
    ! NSPEC for TOT : All reacting species
    integer , public , parameter :: NSPEC_TOT=120
    integer , public , parameter :: NSPEC_ADV=117
    integer , public , parameter :: NSPEC_SHL=3
    integer , public , parameter :: NSPEC_SEMIVOL=23
    integer , public , parameter :: NDRYDEP_ADV = 55
    integer , parameter , public :: NPHOTOLRATES = 17
    integer , parameter , public :: NEMIS_File = 7
    integer , parameter , public :: NEMIS_Specs = 32
```

**Figure 8.** Selected lines from the output file CM_ChemDims_mod.f90 file. (The actual file has comments for each entry.)

```
    integer , public , parameter :: &
       OD          =   1  &
     , OP          =   2  &
     , OH          =   3  &
...
 !+ Defines indices and NSPEC for ADV : Advected species

    integer , public , parameter :: NSPEC_ADV=79
    integer , public , parameter :: FIRST_ADV=29 , &
                                     LAST_ADV=79

    integer , public , parameter :: &
       IXSHL_OD         =   1  &
     , IXSHL_OP         =   2  &
...

    integer , public , parameter :: &
       IXADV_O3         =  29  &
     , IXADV_NO         =  30  &
...
SOMETHING ON   SEMIVOL
```

**Figure 9.** Selected lines from the GenChem-produced file, CM_ChemSpecs_mod.f90.

## 7.2 CM_ChemSpecs_mod.f90

This file specifies basic information about the chemical compounds, in terms of number, indices and some chemical characteristics. Extracts of the file are shown in Fig. 9 and Fig. 10. As seen in Fig. 9, this module provides the simpe indices which
represent the chemical compounds in the EMEP systems, e.g. OH = 3. Indices are additionally provided for the short-lived and advected species (IXSHL_ indices, and IXADV_ indices).

One subroutine, define_chemicals, is generated in this module, which sets the contents of a fortran derived type array named 'species'. The 'species' derived type (c.f. Fig. 10) contains the following elements: name, molwt, nmhc, carbons, nitrogens, and sulphurs. This routine is called in the initialisation of the CTM, and thereafter this array provides useful information on each
420 species, e.g. species(HNO3)%molwt is the molecular weight of HNO3 and species(C5H8)%carbons the number of carbon





```
    !+
    ! Assigns names, mol wts, carbon numbers, advec,  nmhc to user-defined Chemical
    !                                 MW  NM   C   N   S
     species(OD ) = Chemical("OD␣␣",  16.0000,  0,  0,   0,  0 )
...
      species(O3 ) = Chemical("O3␣␣",  48.0000,  0,  0,   0,  0 )
...
   SOME MORE COMPLEX; e.g. SOA

   THINK ABOUT Cstar here vs new GROUPs
```

**Figure 10.** Selected lines from the species array defined in the GenChem-produced file, CM_ChemSpecs_mod.f90

```
   integer, public, target, save, dimension (13) :: &
     RO2_GROUP = (/  &
       HO2, CH3O2, C2H5O2, SECC4H9O2, ISRO2, ETRO2, PRRO2, OXYO2,  &
       MEKO2, MALO2, MVKO2, MACRO2, MACO3  &
     /)
....
   integer, public, target, save, dimension (5) :: &
     WDEP_OXN_GROUP = (/ HNO3, HONO, NO3_f, NO3_c, XHNO3 /)

....
   integer, public, target, save, dimension (2) :: &
     BVOC_GROUP = (/ C5H8, APINENE /)
```

**Figure 11.** Selected lines from the GenChem-produced file, CM_ChemGroups.

atoms in C5H8. The nmhc element identifies if a species is a non-methane hydrocarbon (nmhc=1) or not (nmhc=0). This file also defines pointers to the advected, short-lived and semi-volatile species list, which are used in the EMEP CTM.

### 7.3 CM_ChemGroups_mod.f90

As noted in Sect. 6.1.2, GenChem makes use of the information provided in the Groups column of GenIn_Species.txt to
425 produce fortran arrays, e.g. OXN_GROUP would include all oxidised nitrogen compounds specified with OXN in the Group column (See Fig. 11 for more examples). These groups are also accessible through a fortran pointer system, e.g.

```
    chemgroups(1)%name="RO2"
    chemgroups(1)%ptr=>RO2_GROUP
 ...
    chemgroups(7)%name="DDEP_OXN"
    chemgroups(7)%ptr=>DDEP_OXN_GROUP
```

which allows the EMEP/ESX models to access and perform actions on these groups without having to 'know' the names of the species involved. For example, dry-deposition of all OXN species can be formulated in the model as a simple loop over all





```
...
real , public , target , save , dimension (23) :: &
   DELTAH_GROUP_FACTORS = (/  &
     30.0, 30.0, 30.0, 30.0, 30.0, 30.0, 30.0, 30.0, 30.0, 30.0,  &
     30.0, 30.0, 30.0, 30.0, 30.0, 30.0, 30.0, 30.0, 30.0, 30.0,  &
     112.0, 112.0, 112.0  &
   /)

...
real , public , target , save , dimension (23) :: &
 character(len=TXTLEN_SHORT), public , target , save , dimension (14) :: &
   EXTINC_GROUP_MAPBACK = [ character(len=TXTLEN_SHORT) :: &
   "SO4", "NO3_f", "NO3_c", "NH4_f", "ECn", "ECa", "PMCO",  &
     "ECn", "ECa", "PMCO", "DDf", "DDc", "EC", "DDf"  &
   ]
```

**Figure 12.** Further lines from CM_ChemGroups_mod.f90, illustrating the numerical or string values associated with type (ii) compound groups, see Sect. 6.1.2.

```
integer, public, parameter :: NDRYDEP_ADV = 21
type(depmap), public, dimension(NDRYDEP_ADV), parameter :: DDepMap = (/ &
  depmap(IXADV_O3        , "O3________", -1) &
  ...
, depmap(IXADV_HCHO      , "HCHO______", -1) &
, depmap(IXADV_CH3CHO    , "ALD_______", -1) &
  ...
```

```
integer, public, parameter :: NWETDEP_ADV = 12
type(depmap), public, dimension(NWETDEP_ADV), parameter :: WDepMap = (/ &
  depmap(IXADV_HNO3      , "HNO3______", -1) &
  ...
, depmap(IXADV_NH3       , "NH3_______", -1) &
  ...
```

**Figure 13.** Selected lines from the output file, CM_DryDep.inc, CM_WetDep.inc.

the compounds in either `DDEP_OXN_GROUP_OXN` or, equivalently, by finding the index of `DDEP_OXN` in chemgroups and

following the fortran pointer to the array of indices.

The more complex type (ii) compound groups discussed in Sect. 6.1.2, in which numbers or character strings are associated with a group, result in further arrays in CM_ChemGroup_mod.f90 which provide access to these data. Fig. 12 illustrates some examples of this.

### 7.4   CM_DryDep.inc, CM_WetDep.inc

Species which have had dry or wet-deposition surrogates specified in the 'DRY' or 'WET' column of GenIn_Species.txt are listed as part of a fortran derived type (depmap) in the output files CM_DryDep.inc and CM_WetDep.inc using code such as in Fig. 13, where the first entry is the index of the species in the EMEP/ESX model's list of advected species, and the second





entry is the surrogate species as discussed in Sect. 6.1. The last entry can be used to set fixed values of deposition velocity, although this is not typically done in EMEP/ESX modelling. This listing is then used by the EMEP/ESX models as part of the
445 standard deposition calculations.

### 7.5 CM_Reactions1(2).inc. CM_Reactions.log

Sect. 1.1 has already presented an example section from the file CM_Reactions1.inc. This file generally comprises such production, loss and concetration (P, L, xnew) terms for all, or the majority, of the model's chemically reacting species. For those slowly-reacting species such as $CH4$, a second file CM_Reactions2.inc is produced with the same type of equations, but which
can lie outside of the iteration loop in the EMEP/ESX systems, e.g. the EMEP model code can be summarised as:

```
do iter = 1, NITER
   include 'CM_Reactions1.inc'
end do
include 'CM_Reactions2.inc'
```

The file CM_Reactions.log is not needed by the EMEP or ESX model, but is output as a valuable help-file, containing a
455 listing of all the chemical reactions, along with their index in the 'rct', 'rcphot' and 'rcemis' arrays as appropriate. Equations which were specified with simple constant values (e.g. 1.0e-5) are also reported here with the 'k' indicator and rate.

### 7.6 CM_ChemRates_mod.f90

CM_ChemRates_mod.f90 is the module where all chemical rate-coefficients and photolysis rates are calculated (this is done every advection step in the EMEP code for example). The module is entirely written by GenChem, and produces two subrou-
460 tines:

- setChemRates

- setPhotolUsed

Typical lines of CM_ChemRates_mod.f90 are shown in Fig. 14. This Figure also illustrates how the model makes use of the defined meteorology-associated arrays temp, tinv, rh and Log300divT (Sect. 6.2). The setPhotolUsed array routine is much
simpler, in that it just lists the indices used, e.g.

**Example 7.1.**        *photol_used = (/ IDO3_O3,IDO3_O1D, ... /)*



```
  subroutine setChemRates()

    rct(1,:) = ((5.681e-34*EXP(-2.6*(LOG(TEMP/300))))*O2)*M
    rct(2,:) = (1.8e-11*EXP(107.0*TINV))*N2
....
    rct(65,:) = KAERO(RH)
    rct(66,:) = (IUPAC_TROE(1.0e-31*EXP(1.6*LOG300DIVT),  &
              & 3.0e-11*EXP(-0.3*LOG300DIVT),  &
              & 0.85,  &
              & M,  &
              & 0.75-1.27*LOG10(0.85)))
....
```

**Figure 14.** Selected lines from the output file, CM_ChemRates_mod.f90.

```
integer, parameter, public :: NEMIS_File = 5
character(len=3), save, dimension(NEMIS_File), public :: EMIS_File = (/ &
  "sox" , "nox" , "co␣" , "voc" , "nh3"  /)
```

**Figure 15.** Selected lines from the output file, CM_EmisFile.inc.

## 7.7  CM_EmisFile.inc

The last file simply lists the names used for input emission files. By tradition the EMEP system has used lower case for these
emission markers. The names used are triggered by the 'emisfiles:' line of GenIn_Reactions.txt as discussed in Sect. 6.3. Other
typical emissions that might be used, depending on the application and available inventories are pm25 (fine particulate matter),
or ec, oc, or pom (elemental and organic carbon, or primary organic matter).

## 8  Conclusions

This paper outlines the structure and usage of the GenChem system, which includes a chemical pre-processor (GenChem.py)
for converting chemical equations into differential form for use in atmospheric chemical transport models (CTMs). Although
primarily intended for users of the EMEP CTM and related systems, GenChem also features a simple box-model testing
system (boxChem), which can run as a stand-alone chemical solver, enabling for example easy testing of chemical mechanisms
against each other. GenChem has been developed and tested in a Linux environment, but can be run in virtual environments on
Windows or other architectures.
As provided here, GenChem is already a useful tool to explore different chemical mechanisms, both for gas-phase and
simple (EMEP-compatible) aerosol phase systems. For example, Fig. 3 showed one comparison between the EMEP model's
EmChem19a scheme and the far more advanced CRIv2R5Em and MCMv3.3Em schemes, and Bergström et al. (2020a) provide
many more. Such comparisons will greatly aid the development of new EMEP mechanisms, or indeed for comparison of any
mechanisms of interest to users.



In future, as well as adding new and/or updated chemical mechanisms, we will add utility scripts to simplify result analysis, and to convert between GenChem and other formats, such as those used in KPP (Damian et al., 2002), the MCM website, FACSIMILE (Curtiss and Sweetenham, 1987) or more recently PyBox (Topping et al., 2018).

*Code availability.* The code needed to run the GenChem system is released as open-source code under the GNU license, (https://github.com/metno/genchem), with the user-guide provided at https://genchem.readthedocs.io. The EMEP MSC-W CTM can be found at https:
490   //github.com/metno/emepctm

*Author contributions.* DS wrote the original perl version of the GenChem script, and has written much of the fortran and python code of the more extensive GenChem/boxChem systems. AB wrote the first python version of GenChem.py. RB worked with the chemical mechanisms, HI, JJ worked with and improved many scripts from the GenChem system, including conversion of CRI and MCM codes to GenChem
format. MP added the Docker functionality and helped test the code on Windows. AMV added the Pollen module, and also contributed to some code developments. All authors contributed to the paper.

*Competing interests.* The authors declare that they have no conflict of interest.

*Acknowledgements.* DS was funded by the EU FP7 project ECLAIRE (#282910), and (also for AMV) EMEP under UN-ECE. HI, JJ, RB were funded by the Swedish Strategic Research project MERGE, FORMAS BVOC-SIE project, and (also for MP) Swedish Research Council
project Photosmog project (639-2013-6917). AB was funded by the UK Dept. for Environ., Food and Rural Affairs.





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
