# Peer review of "GenChem v1.0 – a chemical pre-processing and testing system for atmospheric modelling"

_Geoscientific Model Development, 2020_

## Referee Comment (RC1) · Anonymous Referee #1 · 22 Jul 2020

General comments

The paper describes the structure and usage of the GenChem system, which provides scripts and input files for converting chemical equations for use in chemical transport models (CTMs). GenChem was primarily developed for the EMEP model but can be run as a stand-alone chemical solver for testing of chemical mechanisms. One of the main advantages of the system, compared to other such as e.g. KPP, is more human-readable results, which are represented by the name e.g. 'HNO3' instead of numeric or abstract variable representation.

The authors emphasize that the GenChem system can be used as a solver enabling testing of different chemical mechanisms. Several mechanism which are in the system are described in the paper. It is not clear if the system includes mechanisms that are

currently the most popular in chemical transport models. The authors should provide information which mechanisms are the most frequently used in CTMs, recently. Which of these mechanisms are included in GenChem?

The system can be a valuable tool for converting chemical equations to EMEP, which is an open-source model, used by a wider community. It is not clear how about the ESX model. It should be clarified if you can share the code of ESX. Provide the recent applications of this model.

The paper is quite difficult to read, e.g. chapters/subchapters' names are taken from the names of files (e.g. 6.1. GenIn_Shorthands.txt). The names should be more descriptive/should tell what the chapter is about. There are also inconsistencies in use of the models' names, e.g. the same model is called with different names – EMEP MSC-W, EMEP, EMEP CTM, EMEP 3-D CTM, which can make difficult to understand the text for people not familiar with the EMEP model.

Other comments:

- The number of examples of the system application is very limited. The authors often refer to two papers which are in preparation (Bergstrom et al. 2020a and b). I would suggest to add an example of application of e.g. two different chemical mechanisms in EMEP and show the differences in the modelling results.

- In the introduction: include information which are the most popular chemical mechanisms recently used in CTMs and which of these mechanism are in the GenChem system

- Do not use EMEP/ESX system – it should be "EMEP or ESX" or "EMEP and ESX"

- Add information how much time it takes to run the system for different chemical mechanisms and how much time it takes to run the EMEP model with these mechanisms. Precise which of the chemical mechanisms can be used with the EMEP model.

Technical comments

[Figure]

Line 47 – explain what is TWOSTEP, you have not mentioned it before.

Line 70 - "We believe the EMEP model is among the fastest CTMs"– how do you believe that?

Line 128 – options of what?

Line 146 – "if emissions are wanted" – explain what is the difference in using the model with and without emissions included

Line 215 – correct "in ?"

Line 225 – explain what is remPPM

Line 227 – explain what is NPAS

---

## Referee Comment (RC2) · Anonymous Referee #2 · 3 Aug 2020

General comments

This manuscript documents the software package GenChem used to generate the code for calculation of chemical reactions in atmospheric chemical transport models (CTMs). Although in principle being a standalone pre-processor, GenChem is in practice strongly tied to the EMEP CTM. The naming convention, structure etc are closely linked to that model and users without knowledge of the EMEP model could presumably have difficulties getting into the nomenclature. This is not meant as a negative comment. The EMEP CTM is a key model in atmospheric science and European abatement policies and the documentation of GenChem is an important issue in that respect.

Additionally, the possibility of using GenChem for testing and comparing various chem-

ical mechanisms in a box model setup is a very valuable facility.

The manuscript is a technical documentation that appears somewhat like an extended user manual and is somewhat hard to read without actually sitting down and trying out the various parts of the preprocessor. That said, the manuscript seems very well worked through without any major shortcomings and could be accepted with a few minor or technical corrections.

Specific comments

The authors use various names for the EMEP model. It would help the clarity if they stick to one name (e.g. EMEP CTM).

Line 70. It is probably true (as the authors believe) that the EMEP model is among the fastest CTMs, but it would be of to interest to know if there actually has been any comparison of the speed of some of today's main CTMs.

Table 1: The authors might consider to add the output files generated by boxChem either in this table or in a separate table. Now, Table 1 contains the files generated by the GenChem only and not the boxChem.

Technical corrections

Line 105. Replace "box-model code" with boxChem.

Line 150: Species M should be explained here (it is explained later in the doc).

Line 168: Should there be a "-v" in the command similarly to the command at Line 163?

Line 215. Missing chapter info ("?")

Line 227: What does NPAS stand for?

Line 304: "OXN- indicates that this species that belong to a group". Rephrase.

Table 4 (caption). Mentions EMEP's default 'PNAS' scheme. Presumably misspelling

of NPAS?

Suppl. Info (p S6): "Even with dt of 120 s RRMSk values don't exceed 1%.R". Remove the last "R"

---

## Author Comment (AC1) · 30 Sep 2020

**Response to comments from Referees #1 and #2.**

We thank the two referees for their constructive and useful comments. We begin (Section 0) by addressing some queries and comments which were made by both referees, then provide a point by point answer to each referee's comments.

The original referee comments are given in black, and our comments given in blue.

**0. Common Issues**

**0.1 Responses to comments**

Both referees commented that the manuscript could be hard to read.

*Reply: In order to make the paper easier to read we have:*

1. *Modified the headings as suggested by the referee #1*

2. *Harmonised the model naming system, with the more explicit CTM of EMEP MSC-W CTM on first mention, then simply EMEP CTM model.*

3. *Merged some sections to improve the flow:*

   *a) moved and simplified some of the previous TWOSTEP text (old Sect. 1.1) to be part of the main introduction.*

   *b) Merged the boxChem setup texts (start of old Sect. 3 and Sect.4) into one Sect. 3: 'Getting started - GenChem and boxChem basics'.*

   *c) Moved the old Sect.3 text on EMEP CTM preparation into the more explicit new Sect. 4: 'Generating Fortran code for the EMEP CTM model'*

4. *Added a short paragraph at the end of the introduction which describes the structure of the manuscript, and hopefully makes it easier for the reader to focus on the relevant sections, be it overview, installation and usage or mechanistic detail:*

   This paper is mainly intended as a compliment to the user-guide and code provided with GenChem, but we aim to provide here some more discussion of the background and benefits for the approaches chosen. Section 2 focuses on the installation and code structure of the GenChem system. Section 3 illustrates the steps needed to set up and run the boxChem simulations, including plotting commands. This allow users to get a quick-start on the GenChem system, ie to actually run and compare chemical schemes. Section 4 explains how to create and transfer files to the EMEP CTM system. Section 5 explains the many possible options associated with the 'base' and 'extra' chemical mechanisms. Sections 6 explains how to define the chemical mechanisms: detailing the input files which contain chemical species information and reaction mechanisms. Section 7 documents the output files of GenChem, which consist mainly of Fortran code needed for boxChem and EMEP

CTM runs. Finally, Section 8 (Conclusions) discusses some ideas for future development of the GenChem system.

5. *Added some more introductory text at the beginning of some sections*

6. *Greatly shortened the section about the GenIn_Species.csv file (Sect.6.1), since this information was rather too technical, and is available on the readthedocs web-site.*

7. *Generally added some smaller explanatory sentences to improve the flow of the manuscript.*

**0.2 CPU requirements**

The original manuscript stated that 'we believe the EMEP model is among the fastest CTMs', and both referees asked more about this.

*Reply:    The statement was mainly based upon experiences and comments made by others over many years about their CPU requirements. For this reply we have tried to investigate this in a more quantative manner, but find mixed results:*

- *In the EuroDelta multi-model exercise (Bessagnet et al., 2016; Colette et al., 2017), several chemical transport models were run using common meteorology and domains, and a CAMx model was also run with similar but not identical setups. We have been able to compare run times with three of these models, CAMx, CHIMERE, and MATCH (see Bessagnet et al. 2016; Colette et al. 2017; Jiang et al. 2020 for details of models and setup). We found that the EMEP model was several times faster than CHIMERE, somewhat faster than MATCH and slower than CAMx (times from pers. comm. CAMx: Sebnem Aksoyoglu, CHIMERE: Augustin Colette, MATCH: Robert Bergström). However, the CAMx version used a very simple 2-product SOA scheme and 15 vertical layers, CHIMERE used 9 layers, EMEP 20 layers, and MATCH 39 layers, so the CPU times are not straightforward to intepret.*

- *The CAMx webside (www.camx.com/about/speed-scalability.aspx) gives an example of CAMx (v6.40) model performance for a 225×225×25 grid at 12km resolution, using CB6r2 gas-phase chemistry and also various advanced features (e.g plume-in-grid and source apportionment for 9 regions). With 64 cores a walltime of about 20 mins/day is achieved.*

  *For a similar domain (225×225×20) at 0.1 deg resolution, 64 processors, and with CB6r2Em chemistry, the EMEP model uses 2.15 mins/per day. Of course, the CAMx model given in this example uses advanced features such as plume-in-grid modelling, and has more vertical levels, so again it is hard to make a consistent comparison.*

- *Delic (2018) investigated the MPI performace of the CMAQ model, using a 24h test-case, for a grid of $100 \times 80$ California domain of 12 km resolution, with 35 vertical layers. They used a chemical mechanism of 149 active species and 329 reactions. Their test case required 16.7 mins per day using 32 processors (with an MPI efficiency of 0.63), or 10.8 mins/day with 64 processors (efficiency of 0.49). Although we cannot perform identical tests with the EMEP system, we have set up a simulation which should be reasonably comparable: a $100 \times 82$ European domain of $0.1 \times 0.1$ lat./lon. (ca. 10 km) resolution, with 35 vertical layers. As the EMEP model does a lot of pre-processing to interpolate e.g. emissions and landcover to the in-use grid on the first time step, the CPU times of a 1-day simulation are not representative of the typical time needed. We therefore tested a 31-day simulation to get average run-times for 24h also. With EmChem19a the EMEP model requires 0.74 mins/day from a 1-day run, or 0.48 mins/day from a 31-day run, using 64 processors. With CB6r2Em these runs take 0.85 mins/day and 0.6 mins/day.*

*In summary, it is difficult to ensure comparability of many factors, including model setup, computing platforms and usage (number of processors, etc), so we are reluctant to publish CPU numbers from these different comparisons. We have also simplified the text surrounding TWOSTEP in the introduction, and found it best to simply omit any attempt to compare with other models. We do however add explicit CPU time for EMEP CTM runs in Table 3 (new numbering).*

**0.3 NPAS**

Line 227 explain what is NPAS

*Reply: We have added text to explain that NPAS means no-partitioning of primary organic matter, and with aging of secondary organics. Further details can be found in the cited reference of Simpson et al. 2012.*

**0.4 Other changes**

A number of changes have been made in the GenChem system since publication of the original manuscript as a Discussions paper:

- The file 'GenIn_Shorthands.txt' which used to reside the chem/scripts directory, and which serves as the initial default set of shorthands, has now been moved and renamed as chem/generic_Shorthands.txt, in order to make it more visible to users.

- The logarithms used for CB6 in the Shorthands file should have been log10 rather than natural logarithms. This bug has been fixed.

- The emissions speciation file for CRIv2R5Em has been modified to bring it into line with the version used in the Bergström et al. (2020a) paper.

- The boxChem script box/scripts/getboxconcs.py has been updated so that its arguments are more similar to boxplots.py; with -v and -i, following the comment of Ref.#2.

- The boxChem script do.testChems now uses the default output directory OUTPUTS, see the new Table 2.

As a final comment on the code, we can note that the github code is currently tagged as 1.0.0-beta. If the resubmitted manuscript is accepted this code will be re-tagged as 1.0 when the manuscript is published.

**1 Referee #1**

**1.1 General comments**

The authors emphasize that the GenChem system can be used as a solver enabling testing of different chemical mechanisms. Several mechanism which are in the system are described in the paper. It is not clear if the system includes mechanisms that are currently the most popular in chemical transport models. The authors should provide information which mechanisms are the most frequently used in CTMs, recently. Which of these mechanisms are included in GenChem?

*Reply: The intention of this paper is to focus on GenChem as a tool and not too much on the actual chemical mechanisms, but we have added some words to the conclusions of the manuscript:*

*The mechanisms included now reflect those used or made available for the EMEP CTM, as well as the MCM scheme which works in the boxChem mode. The EmChem19a scheme is only used in the EMEP CTM, but we include slightly adapted versions of CB6 which is used in the widely used CAMx model (http://www.camx.com) or CMAQ (Luecken et al., 2019), and CRIv2-R5 scheme which is used in STOCHEM (Archibald et al., 2010; Khan et al., 2015). It is hoped that some of the other widely-used mechanisms can be added in future, for example the MOZART scheme (Emmons et al., 2010; Surendran et al., 2015), the RACM scheme (Stockwell et al. (1997); Goliff et al. (2013)), or SAPRC-07 (Carter, 2010) which is also used in CMAQ (https://www.airqualitymodeling.org/index.php/CMAQv5.1_Mechanisms).*

The system can be a valuable tool for converting chemical equations to EMEP, which is an open-source model, used by a wider community. It is not clear how about the ESX model. It should be clarified if you can share the code of ESX. Provide the recent applications of this model.

*Reply: The ESX model should indeed be released as an open-source model, but we decided we had to finish and release the GenChem and updated EMEP CTM systems first. The only published information on ESX is the EMEP chapter which was cited, and the final report (which seems unaccessible now) of the EU ECLAIRE project which initiated the work. Since then a lot of work has been done with the ESX model, and a proper documentation is needed. Looking over the GenChem manuscript now we see that ESX is indeed mentioned many times. In view of its in-preparation status, we have reduced the number of references to the ESX model in the revised manuscript. (We hope to release ESX as open-source on github in the next months, but some preparatory work and final-checks are needed prior to this.)*

The paper is quite difficult to read, e.g. chapters/subchapters names are taken from the names of files (e.g. 6.1. GenIn_Shorthands.txt). The names should be more descriptive/should tell what the chapter is about. There are also inconsistencies in use of the models names, e.g. the same model is called with different names EMEP MSC-W, EMEP, EMEP CTM, EMEP 3-D CTM, which can make difficult to understand the text for people not familiar with the EMEP model.

*Reply: Please see comments given in Section 0.1 above.*

**1.2    Other comments**

- The number of examples of the system application is very limited. The authors often refer to two papers which are in preparation (Bergstrom et al. 2020a and b). I would suggest to add an example of application of e.g. two different chemical mechanisms in EMEP and show the differences in the modelling results.

*Reply:   As noted above, the intention of this paper is to focus on GenChem as a tool and not too much on the actual chemical mechanisms. The Bergström et al 2020a paper compares the mechanisms in detail, and presents both boxChem and EMEP CTM results for the different mechanisms. This paper is in its final stages of preparation and will shortly be submitted to GMD. We think it would confuse the intention of the current GenChem paper if we start comparisons of the actual mechanisms.*

- In the introduction: include information which are the most popular chemical mechanisms recently used in CTMs and which of these mechanism are in the GenChem system.

*Reply: As noted in the reply to the referee given above, we have now mentioned other popular schemes (MOZART, CAM-4, RADM, SAPRC) in the conclusions section, and identified these as candidates for future inclusion in the GenChem system.*

- Do not use EMEP/ESX system  it should be "EMEP or ESX or "EMEP and ESX

*Reply: We have changed the text as requested, and indeed removed many mentions of ESX*

- Add information how much time it takes to run the system for different chemical mechanisms and how much time it takes to run the EMEP model with these mechanisms. Precise which of the chemical mechanisms can be used with the EMEP model.

*Reply: We have extended Table 2 (now Table 3) with times for boxChem and EMEP model runs, and made it clear that all mechanisms except MCMv3.3Em can be used in the EMEP model. Further examples will be presented in Bergström et al. (2020a)*

**1.3   Technical comments**

Line 47  explain what is TWOSTEP, you have not mentioned it before.

*Reply: We have added the simple '(see below)' on first mention in the bullet points, but then brought forward a short explanation of TWOSTEP. As noted in Sect. 0.1 above, the original subsection on TWOSTEP has now been shortened and brought into the main introduction text.*

Line 70 - "We believe the EMEP model is among the fastest CTMs" how do you believe that?

*Reply: Please see comments given in Section 0.2 above.*

Line 128  options of what?

*Reply: We have changed 'list of available options' to be more explicit. These options are a usage message, a debug flag, and a list of available chemical mechanisms, and a debug option.*

Line 146  "if emissions are wanted"  explain what is the difference in using the model with and without emissions included

*Reply: We have modified the text as follows:*

By default, boxChem uses the set of emission rates as specified by variables set in config_box.nml, currently set with the lines beginning:

`emis_kgm2day = 'nox', 18.3 !  NOx, kg/m2/day,`

with 'voc' emissions set on the next line as 15.4 kg/m2/day. These emissions are converted by boxChem to instantaneous production rates in molecules $cm^{-3}$ $s^{-1}$, accounting for molecular masses, emissions speciation (e.g. nox as NO and $NO_2$) and the mixing height, Hmix (also set in config_box.nml). Such emission rates can be modified by the user, or indeed all emissions set to zero if the variable use_emis is set to 'F' (False).

Line 215  correct "in ?"

*Reply: This should have been Bergström et al. 2020b*

Line 225  explain what is remPPM

*Reply: We have added text to explain that this is the remaining PPM component.*

Line 227  explain what is NPAS

*Reply: Please see reply above, in Sect.0.3*

**2 Referee #2**

**2.1 General comments**

This manuscript documents the software package GenChem used to generate the code for calculation of chemical reactions in atmospheric chemical transport models (CTMs). Although in principle being a standalone pre-processor, GenChem is in practice strongly tied to the EMEP CTM. The naming convention, structure etc are closely linked to that model and users without knowledge of the EMEP model could presumably have difficulties getting into the nomenclature. This is not meant as a negative comment. The EMEP CTM is a key model in atmospheric science and European abatement policies and the documentation of GenChem is an important issue in that respect.

*Reply: Yes, these are fair comments.*

Additionally, the possibility of using GenChem for testing and comparing various chemical mechanisms in a box model setup is a very valuable facility. The manuscript is a technical documentation that appears somewhat like an extended user manual and is somewhat hard to read without actually sitting down and trying out the various parts of the pre-processor. That said, the manuscript seems very well worked through without any major shortcomings and could be accepted with a few minor or technical corrections.

*Reply: Yes, this is also a fair summary. As noted in the reply to Ref.#1 we have tried to make the manuscript easier to read in some respects, but it remains quite technical due to the nature of the system description.*

**2.2 Specific comments**

The authors use various names for the EMEP model. It would help the clarity if they stick to one name (e.g. EMEP CTM).

*Reply: We have simplified this, referring to the EMEP MSC-W CTM on first use, and thereafter just EMEP CTM.*

Line 70. It is probably true (as the authors believe) that the EMEP model is among the fastest CTMs, but it would be of to interest to know if there actually has been any comparison of the speed of some of todays main CTMs.

*Reply: Please see comments given in Section 0.2 above.*

Table 1: The authors might consider to add the output files generated by boxChem either in this table or in a separate table. Now, Table 1 contains the files generated by the GenChem only and not the boxChem.

*Reply: We have modified the caption and column headers for Table 1 to make it clear what that table refers to. We have added a new table (Table 3, new numbering) with the output files from do.testChem runs. We also modified the do.testChems script so that the log file (eg RES.EmChem19a) is also written into this directory.*

**2.3   Technical corrections**

Line 105. Replace box-model code with boxChem.

*Reply: We have made this change.*

Line 150: Species M should be explained here (it is explained later in the doc).

*Reply: We have added a reference to the Sect. 6.2 where this is explained.*

Line 168: Should there be a -v in the command similarly to the command at Line 163?

*Reply: The script getboxconcs.py had different arguments, so the line as given was correct. However, as noted in Sect. 0.4 above, we have now modified the script so that its arguments are more similar to boxplots.py; with -v and -i.*

Line 215. Missing chapter info (?)

*Reply: This should have been the in-preparation manuscript of Bergström et al. 2020b*

Line 227: What does NPAS stand for?

*Reply: Please see reply above, in Sect.0.3*

Line 304: OXN- indicates that this species that belong to a group. Rephrase.

*Reply: Done.*

Table 4 (caption). Mentions EMEPs default PNAS scheme. Presumably misspelling of NPAS?

*Reply: Corrected. Yes, this should have been NPAS.*

Suppl. Info (p S6): Even with dt of 120 s RRMSk values dont exceed 1

*Reply: Done.*

**References**

Archibald, A. T., Cooke, M. C., Utembe, S. R., Shallcross, D. E., Derwent, R. G., and Jenkin, M. E.: Impacts of mechanistic changes on HOx formation and recycling in the oxidation of isoprene, Atmos. Chem. Physics, 10, 8097–8118, 2010.

Bergström, R., Jenkin, M., Hayman, G., and Simpson, D.: Update and comparison of atmospheric chemistry mechanisms for the EMEP MSC-W model system — EmChem19a, EmChem19X, CRIv2R5Em, CB6r2Em, and MCMv3.3Em, To be submitted, 2020a.

Bergström, R., et al.: Organic aerosol schemes for the EMEP MSC-W model for European and Global scale simulations, In preparation, 2020b.

Bessagnet, B., Pirovano, G., Mircea, M., Cuvelier, C., Aulinger, A., Calori, G., Ciarelli, G., Manders, A., Stern, R., Tsyro, S., García Vivanco, M., Thunis, P., Pay, M.-T., Colette, A., Couvidat, F., Meleux, F., Rouïl, L., Ung, A., Aksoyoglu, S., Baldasano, J. M., Bieser, J., Briganti, G., Cappelletti, A., D'Isidoro, M., Finardi, S., Kranenburg, R., Silibello, C., Carnevale, C., Aas, W., Dupont, J.-C., Fagerli, H., Gonzalez, L., Menut, L., Prévôt, A. S. H., Roberts, P., and White, L.: Presentation of the EURODELTA III intercomparison exercise – evaluation of the chemistry transport models' performance on criteria pollutants and joint analysis with meteorology, Atmospheric Chemistry and Physics, 16, 12 667–12 701, https://doi.org/10.5194/acp-16-12667-2016, URL `http://www.atmos-chem-phys.net/16/12667/2016/`, 2016.

Carter, W. P.: Development of the SAPRC-07 chemical mechanism, Atmos. Environ., 44, 5324 – 5335, https://doi.org/https://doi.org/10.1016/j.atmosenv.2010.01.026, URL `http://www.sciencedirect.com/science/article/pii/S1352231010000646`, atmospheric Chemical Mechanisms: Selected Papers from the 2008 Conference, 2010.

Colette, A., Andersson, C., Manders, A., Mar, K., Mircea, M., Pay, M.-T., Raffort, V., Tsyro, S., Cuvelier, C., Adani, M., Bessagnet, B., Bergström, R., Briganti, G., Butler, T., Cappelletti, A., Couvidat, F., D'Isidoro, M., Doumbia, T., Fagerli, H., Granier, C., Heyes, C., Klimont, Z., Ojha, N., Otero, N., Schaap, M., Sindelarova, K., Stegehuis, A. I., Roustan, Y., Vautard, R., van Meijgaard, E., Vivanco, M. G., and Wind, P.: EURODELTA-Trends, a multi-model experiment of air quality hindcast in Europe over 1990–2010, Geoscientific Model Dev., 10, 3255–3276, https://doi.org/10.5194/gmd-10-3255-2017, URL `https://gmd.copernicus.org/articles/10/3255/2017/`, 2017.

Delic, G.: CMAQ 5.2.1 Parallel performance with MPI and OPENMP, URL `https://www.researchgate.net/publication/328074959`, presented at the 17 th Annual CMAS Conference, Chapel Hill, NC, October 22-24, 2018, 2018.

Emmons, L. K., Walters, S., Hess, P. G., Lamarque, J.-F., Pfister, G. G., Fillmore, D., Granier, C., Guenther, A., Kinnison, D., Laepple, T., Orlando, J., Tie, X., Tyndall, G., Wiedinmyer, C., Baughcum, S. L., and Kloster, S.: Description and evaluation of the Model for Ozone and Related chemical Tracers, version 4 (MOZART-4), Geoscientific Model Dev., 3, 43–67, https://doi.org/10.5194/gmd-3-43-2010, URL https://gmd.copernicus.org/articles/3/43/2010/, 2010.

Goliff, W. S., Stockwell, W. R., and Lawson, C. V.: The regional atmospheric chemistry mechanism, version 2, Atmos. Environ., 68, 174 – 185, https://doi.org/http://dx.doi.org/10.1016/j.atmosenv.2012.11.038, URL http://www.sciencedirect.com/science/article/pii/S1352231012011065, 2013.

Jiang, J., Aksoyoglu, S., Ciarelli, G., Baltensperger, U., and Prvt, A. S.: Changes in ozone and $PM_{2.5}$ in Europe during the period of 19902030: Role of reductions in land and ship emissions, Sci. of the Total Environ., 741, 140 467, https://doi.org/https://doi.org/10.1016/j.scitotenv.2020.140467, URL http://www.sciencedirect.com/science/article/pii/S0048969720339899, 2020.

Khan, M. A. H., Cooke, M. C., Utembe, S. R., Archibald, A. T., Derwent, R. G., Jenkin, M. E., Morris, W. C., South, N., Hansen, J. C., Francisco, J. S., Percival, C. J., and Shallcross, D. E.: Global analysis of peroxy radicals and peroxy radical-water complexation using the STOCHEM-CRI global chemistry and transport model, Atmos. Environ., 106, 278–287, https://doi.org/{10.1016/j.atmosenv.2015.02.020}, 2015.

Luecken, D., Yarwood, G., and Hutzell, W.: Multipollutant modeling of ozone, reactive nitrogen and HAPs across the continental US with CMAQ-CB6, Atmospheric Environment, 201, 62 – 72, https://doi.org/https://doi.org/10.1016/j.atmosenv.2018.11.060, URL http://www.sciencedirect.com/science/article/pii/S1352231018308434, 2019.

Stockwell, W., F.Kirchner, and M.Kuhn: A New Mechanism for Regional Atmospheric Chemistry Modeling, J. Geophys. Res., 102, 25 847–25 879, 1997.

Surendran, D. E., Ghude, S. D., Beig, G., Emmons, L., Jena, C., Kumar, R., Pfister, G., and Chate, D.: Air quality simulation over South Asia using Hemispheric Transport of Air Pollution version-2 (HTAP-v2) emission inventory and Model for Ozone and Related chemical Tracers (MOZART-4), Atmospheric Environment, 122, 357 – 372, https://doi.org/http://dx.doi.org/10.1016/j.atmosenv.2015.08.023, URL http://www.sciencedirect.com/science/article/pii/S1352231015302685, 2015.

---

## Author Response (AR2)

Dear Editor,

Please find attached final versions of the manuscript gmd-2020-147 (which was accepted 'as-is'), GenChem v1.0 a chemical pre-processing and testing system for atmospheric modellin. I include both pdf and latex versions, along with figures and supplementary information.

With Best Regards,

David Simpson, 26/10/2020